# Lactose Intolerance and Bone Health: The Challenge of Ensuring Adequate Calcium Intake

**DOI:** 10.3390/nu11040718

**Published:** 2019-03-28

**Authors:** Joanna K. Hodges, Sisi Cao, Dennis P. Cladis, Connie M. Weaver

**Affiliations:** Department of Nutrition Science, Purdue University, West Lafayette, IN 47907, USA; cao226@purdue.edu (S.C.); dcladis@purdue.edu (D.P.C.); weavercm@purdue.edu (C.M.W.)

**Keywords:** bone, calcium, lactose intolerance, osteoporosis

## Abstract

Calcium is an important nutrient with impact upon many biological systems, most notably bone. Ensuring adequate calcium intake throughout the lifespan is essential to building and maintaining bone. Lactose intolerance may predispose individuals to low calcium intake as the number of lactose-free, calcium-rich food sources is limited. In this review, we summarize data from human and animal studies on the influence of lactose and lactase deficiency on calcium absorption and bone health. Based on the available evidence, neither dietary lactose nor lactase deficiency have a significant impact on calcium absorption in adult humans. However, lactose intolerance may lead to reduced bone density and fragility fractures when accompanied by decreased intake or avoidance of dairy. Recently published human trials and meta-analyses suggest a weak but significant association between dairy consumption and bone health, particularly in children. Given the availability of simple dietary approaches to building lactose tolerance and the nutritional deficiencies associated with dairy avoidance, multiple public health organizations recommend that all individuals—including those that are lactose intolerant—consume three servings of dairy per day to ensure adequate nutrient intakes and optimal bone health.

## 1. Bone Health throughout the Life Span

Bone is a dynamic tissue that is constantly changing throughout the lifetime. As shown in Figure 1, bone accrues rapidly during childhood and adolescence, stays relatively steady in early adulthood, and resorbs in later adulthood, potentially leading to osteoporosis. Osteoporosis constitutes a major health burden, with an average of 2 million osteoporotic fractures annually [1]. These fractures are especially common in females over age 55, accounting for 40% of all hospitalizations and related costs, which is nearly as much as myocardial infarction, stoke, and breast cancer combined [1]. Osteoporotic fractures lead to a lower quality of life and many individuals suffering these fractures cannot live independently again [2,3]. Given the extensive and costly impact of osteoporosis, taking steps throughout life to prevent it is essential to lowering this health burden. As dietary calcium is one of the primary strategies to build and maintain strong and healthy bones, it is critical to ensure adequate calcium intake at each life stage, especially in lactose-intolerant individuals that consume less dietary calcium than their peers [4].

*Early years: bone modeling.* Growing children and adolescents form new bone (termed bone modeling) at a rapid rate. Bone modeling requires adequate calcium to properly form and lay down new bone. This is especially true during the pubertal growth spurt, because one-third of total bone mass is accrued between Tanner stages 2–4, with an average calcium accretion rate of 300–400 mg/d [5]. As approximately 95% of adult peak bone mass is acquired by 16 years of age, ensuring adequate calcium intake during this time is key to bone health [6].

*Middle years: bone remodeling.* After the pubertal growth spurt, only small gains in net bone mass are possible because of the closure of the epiphyseal plates. During this phase of life, old bone is resorbed by osteoclasts and is replaced by new bone formed by osteoblasts. This process is called remodeling, as there is no net change in bone mass. In healthy adults, the entire skeleton will be remodeled every 10 years [7]. Thus, continually ensuring adequate calcium intake during the adult years is necessary to maintain peak bone mass.

*Later years: bone resorption.* In aging adults, the bone remodeling process becomes uncoupled, with bone resorbing osteoclasts being more active than bone forming osteoblasts. This results in lowered bone mass and increased susceptibility to fragility fractures. Females are most vulnerable to osteoporotic fractures, with half of all females over 50 years of age suffering an osteoporotic fracture (compared to 1 in 5 men) [8]. This is largely due to the 7–10 years surrounding menopause, as females lose an average of 3–7% of total bone mass annually during this time [9]. Although bone loss slows to 1–2% after age 65 (comparable to elderly males), much of the damage has already been done, leaving females at a higher risk of fracture [10,11].

## 2. Calcium Requirements

The primary nutrient of interest in bone health is calcium. A large body of evidence is available concerning the effects of dietary calcium throughout the lifespan. The National Academies of Medicine (NAM, formerly the Institute of Medicine (IOM)) regularly review and synthesize this evidence to form Dietary Reference Intakes (DRIs). In the latest update, NAM considered the evidence for a number of different health endpoints, but concluded that the only convincing evidence for setting calcium DRIs is on the basis of bone health [12]. The Recommended Dietary Allowances (RDAs) set by NAM are shown in Table 1.

In setting the recommendations for calcium intake, NAM relied primarily on calcium balance studies for persons up to 50 years of age [12]. As seen in Table 1, calcium requirements are the same for males and females during the first 50 years of life, with recommended intakes being highest during the adolescent years when maximal bone growth occurs. No evidence was found to suggest increasing calcium intakes during pregnancy and lactation, thus, recommended intakes during these periods are no different from age-matched peers [12]. In later years, the RDA for females increases earlier than for males to mitigate the effects of menopause on bone loss, though the recommended intake for both sexes increases at age 70 to preserve bone and prevent the development of osteoporosis [12].

Across all age groups, average calcium intakes fail to meet the RDAs [12]. To remedy this, many individuals turn to calcium-containing dietary supplements to help meet their calcium needs [13]. While this strategy can be beneficial, those consuming dietary supplements must take care not to exceed the Upper Limit (UL). Calcium intakes exceeding the UL increase the risk of adverse events, including hypercalcemia, hypercalciuria, vascular and soft tissue calcification, kidney stones, and constipation [12]. This is particularly concerning in adult females, as <5% have calcium intakes above the current UL [12]. For lactose-intolerant individuals and other non-dairy consumers, dietary supplements offer a way to meet recommended intakes for bone health, though these individuals must closely monitor the amount consumed to avoid exceeding the UL.

## 3. Dietary Sources of Calcium

To meet the calcium recommendations set forth by NAM, consuming a variety of calcium-rich foods each day is necessary. Historically, dairy products were the predominant source of dietary calcium, accounting for 70% of calcium intake in the American diet as recently as 1984 [14]. Today, however, dairy products account for a mere 40% of calcium intake [15], largely due to the popularity of dietary supplements and the declining intake of cow’s milk. In fact, calcium-containing dietary supplements are consumed by 40% of adults and 70% of elderly persons regularly [16].

Although dietary habits have shifted in recent years, consuming a diverse diet with a variety of calcium-rich sources is still essential to meeting the calcium recommendations set forth by NAM [12]. Dairy products (e.g., milk, yogurt, and cheese) provide the most concentrated sources of calcium in the diet (Table 2). In addition, food companies have developed a number of dairy-substitute drinks fortified with calcium (Table 2). However, outside of dairy and these beverages, there are few calcium-rich foods for non-dairy consumers, most of which contain less calcium per serving than dairy products and dairy substitutes (Table 3). Many plant-based foods also contain oxalates and phytates, which decrease the bioavailability of calcium [17]. Thus, it can be difficult for consumers who avoid dairy to achieve adequate calcium intakes.

## 4. Lactose Intolerance and Calcium Bioavailability from Dairy and Alternative Sources

The majority of calcium absorption studies show that neither dietary lactose nor lactase deficiency in healthy adults has a significant impact on calcium absorption. While the presence of lactose may stimulate calcium absorption in animals [19] and infants [20], this effect does not persist into adulthood in humans [21,22], nor is calcium absorption affected by lactase deficiency in adults [23,24,25]. A study by Nickel et al. [26] found no significant difference in calcium absorption from a range of dairy products differing in lactose level (milk, yogurt, cheddar cheese, processed cheese) in adult women. Similarly, using a single calcium isotope, Horowitz [27] found no relationship between lactose content and calcium absorption and an equal absorption efficiency in postmenopausal women, whether or not they were maldigesters. The lack of effect of lactose on calcium absorption may not hold true in the elderly, although evidence to support this is ambiguous. Schuette et al. [28] used a single isotope calcium test and found that 12 g of lactose added to a non-carbohydrate milk formula enhanced calcium absorption in postmenopausal women. On the other hand, Obermayer-Pietsch et al. [29] showed that calcium absorption in postmenopausal women with a genetic predisposition to lactose maldigestion was 56% lower from water containing 50 g of lactose compared to water alone. Therefore, the effect may depend on the specific type of food source and the dose of lactose.

As in the case of lactose content in food, lactase deficiency in the consumer appears not to be a limiting factor for calcium absorption. Using an extrinsic tracer, Smith et al. [23] found similar absorption of calcium from milk and yogurt in individuals with and without lactase deficiency. Using a double-isotope method, Tremaine et al. [24] compared calcium absorption between lactose containing milk and hydrolyzed milk in lactase-deficient and sufficient adults and found that calcium was absorbed equally well regardless of the source in participants with lactase deficiency. Interestingly, the lactase deficient participants had a higher calcium absorption efficiency compared to control group. This finding was similar to that by Griessen et al. [25], who used a parallel study design in young adults to show that participants with lactase deficiency tended to absorb calcium more efficiently than controls from drinks with and without lactose. These results may stem from either the habitually lower calcium intake of participants with lactose intolerance or the bifidogenic effect of unabsorbed lactose [30].

## 5. Calcium Bioavailability from Alternative Sources and the Impact on Bone Health

Plant-based beverages, such as those made out of ground rice, almonds, soy nuts, and oats, are becoming increasingly popular among consumers as substitutes for cow’s milk [31]. They are also advertised as better-for-you alternatives [32]. According to consumer trend reports, 49% of Americans, including 68% of parents and 54% of children, consume these beverages for their perceived health benefits [33,34]. Most plant-based beverages are supplemented with an equal or greater amount of calcium and vitamin D compared to that in cow’s milk [32,35]. However, the bioavailability of these nutrients from non-dairy drinks is largely unknown, except for soy drink, which is the most common milk substitute and the first one to appear as such on the U.S. market [36].

Plant-based drinks contain higher levels of calcium and zinc absorption inhibitors, such as phytate and oxalate. In general, calcium absorption is inversely proportional to the content of these compounds. However, soy beans, despite being rich in both phytates and oxalates, show relatively high calcium bioavailability. Experiments with dual-isotope calcium tracers showed that calcium absorption from fortified soy drink was equivalent to that from cow’s milk (20–40% depending on the calcium status of the subject [37]), but only if calcium carbonate was used as the fortificant [38]. Absorption from tricalcium phosphate-fortified beverage was 20% lower, possibly due to the intestinal precipitation of calcium by phosphate. However, a study that employed a single isotope method to measure calcium absorption at a lower dose of calcium (44 mg vs. 250 mg) and over a shorter time period (1 h vs. 24 h) found that the absorption of calcium from soy drink fortified with calcium phosphate was equivalent to that from cow’s milk [39]. The discrepancy may be explained by changes in absorbability due to bacterial fermentation in the lower gut, which requires a longer study period for detection.

Another type of beverage frequently investigated for calcium bioavailability is calcium-fortified orange juice. Similarly to soy drink, the absorption of calcium from orange juice was found to be comparable to that from cow’s milk when measured by a single isotope test in adult men and women [40]. The absorption of calcium was also compared between two fortification methods of orange juice: calcium citrate malate and a combination of tricalcium phosphate and calcium lactate [41]. The investigators found that calcium citrate malate was superior in terms of calcium bioavailability. These findings indicate that calcium absorption depends not only on the physical formulation of the product and the experimental design (length of time over which absorption is measured) but also the fortification method. Therefore, equivalent calcium contents on a nutritional label do not guarantee equivalent nutritional value [41]. Bioavailability must be measured directly using standardized and generalizable methods [42].

To date, there are no data on the bioavailability of calcium from plant-based drinks other than soy drink, despite their increasing popularity. Nor is there any knowledge about the bioavailability of other bone-related nutrients, such as protein. Most plant-based beverages, except soy drink, are 5–10 times lower in protein [32,35]. Protein is important for bone because it enhances insulin-like growth factor 1 that exerts positive activity on skeletal development and bone formation [43]. A continuous supply of protein is also required for optimal bone remodeling to provide the amino-acids that were removed during bone resorption [44]. A recent summary of meta-analyses suggests that dietary protein is just as important as adequate calcium intake in reducing bone loss and the risk of hip fracture [45].

The contents of other nutrients important to bone, including potassium, phosphorus, and magnesium, vary from one plant-based drink to another. For example, almond drink has a similar magnesium content to that of cow’s milk but is higher in sodium, which is negatively associated with bone health [32,35]. Coconut drink is lower in sodium and similar in magnesium to cow’s milk, while cashew drink is both higher in sodium and lower in magnesium. The nutrient profiles of several popular cow’s milk alternatives are shown in Table 4. As in the case of calcium, the bioavailability of these nutrients from plant-based drinks has not been established. More randomized-controlled trials are needed to determine whether they can be used to replace cow’s milk from diet without causing nutrient deficiencies.

Despite the increasing popularity of plant-based beverages, a considerable gap in knowledge also exists with regard to their impact on bone mineral content (BMC) and the risk of fracture. Similar to research on calcium bioavailability, most studies examining bone health focused on soy drink. A study by Gui et al. [46] found that postmenopausal Chinese women consuming cow’s milk for 18 months had a higher bone mineral density (BMD) in the hip and femoral neck compared to women consuming soy drink. On the other hand, the association between soy milk versus dairy milk consumption and the risk of osteoporosis was no different in postmenopausal white women [47]. However, bone density in this study was assessed in heel bone only, which may explain the discrepancy in findings. The effects may also differ depending on the age of the target population. In a longitudinal study of healthy infants, Andres et al. [48] found that bone mineral accretion during the first year of life was higher in infants fed soy formula compared to those fed breast milk or cow’s milk formula. However, since the majority of lactase deficiencies in infants are transient and due to diarrhea, the use of low-lactose or lactose-free formulas has no clinical advantage, except in severely undernourished infants with diarrhea, in whom lactose-free formula may be advantageous [49].

## 6. The Effect of Lactose on Calcium Metabolism and Bone Health in Animal Studies

The in vivo effects of lactose on calcium absorption, bone growth, and BMC were extensively investigated in the 1980s and 1990s. Lactose was shown to enhance calcium bioavailability from a variety of sources at different life stages of development [50,51]. There are two pathways of calcium transport considered in the mechanism of lactose-promoted calcium absorption: passive transport through all segments of the small intestine and facilitated diffusion in the jejunum. It has been reported that lactose enhanced calcium absorption by increasing calcium permeability in small intestinal villi [51] and lowering the ileal pH in rats [52]. In addition to calcium absorption, lactose, together with calcium, significantly improved recalcification of bones in calcium-deficient rats [53]. Although Shortt et al. [54] reported that lactose alone did not seem to alter bone mass or bone breaking strength in rats, and they demonstrated that lactose protected against the high salt intake-induced bone loss in weaning rats by enhancing calcium absorption. Moreover, in two trials using post-weaning rats and swine conducted by Moser et al. [55], bone breaking strength was not altered by dietary lactose. However, in the post-weaning rat model, higher levels of lactose in the diet corresponded to increased skeletal calcium content.

Numerous studies demonstrate that dietary lactose facilitates intestinal calcium absorption and promotes skeletal growth independent of the vitamin D endocrine system. Miller et al. [56] showed that, by feeding weaning rats with 20% lactose supplementation in vitamin D-deficient diets for 4 weeks, bone weights, bone calcium, and endochondral bone elongation were significantly improved in the lactose-fed rats compared to sucrose-fed controls. The changes observed in the lactose-supplemented group were similar to the vitamin D-replete group. Similar results were reported by Au et al. [57] and Schaafsma et al. [58] by feeding weaning, vitamin D-deficient rats with lactose or calcium-lactose supplemented diets, respectively, for 6 weeks.

In addition to pure lactose, milk containing lactose also exerted a better bone protective effect when compared to lactose-hydrolyzed milk [59] or calcium carbonate [60]. It has been reported that weanling male rats fed the untreated milk for 28 days absorbed calcium more efficiently and showed reduced urinary calcium loss compared to the rats fed with lactose-hydrolyzed milk. Moreover, the rats fed with unhydrolyzed milk or supplemental lactose retained more magnesium and zinc in bone [59]. Weaver et al. [60] also demonstrated that in a growing rat model, bones were larger and stronger when the rats were fed adequate calcium diets, with the calcium sourced from nonfat dry milk compared to that from calcium carbonate. In addition, rats fed with nonfat dry milk during growth retained higher bone density and strength in adulthood when the diets were switched to the same low calcium diet compared to rats fed with calcium carbonate.

## 7. Human Studies of Lactose Intolerance, Dairy Avoidance, and Bone Health

Lactose maldigestion, when combined with inadequate calcium intake, has been suggested as a risk factor for impaired bone health. Multiple observational studies show that consumers who avoid milk have lower BMD compared to individuals with higher milk consumption. This association is particularly evident in children [61,62,63,64,65]. Observational and longitudinal studies conducted in Europe [63], Asia [66], and New Zealand [64] demonstrated that children with lactose-free or low lactose diets had reduced BMC and BMD. Another study of early pubertal girls in California and Indiana found that perceived milk intolerance was inversely associated with BMC of several bone sites [67]. A meta-analysis of trials of dairy products and dietary calcium on BMC in children showed significantly higher total body and lumbar spine BMC in groups with higher compared to low calcium intakes [61]. Several randomized controlled trials also found that increasing lactose intake through a dairy intervention in children resulted in improved BMC, particularly in girls with low baseline calcium intake [65,68,69,70].

In adolescents, a study by Baldan et al. demonstrated that lactose malabsorbers who substituted regular cow’s milk with lactose-free milk showed no difference in bone mineral status as compared to controls, despite a lower calcium intake [4]. In adults, BMD was found to be lower with lactose maldigestion and lower dairy intakes [71,72,73]. A recently published population-based study of 2040 elderly men and women also found a weak but significant positive association between dairy product consumption and bone properties [74]. However, inconsistencies in adjustment for confounders and the varying magnitude of association found in previous studies preclude firm conclusions [65,75,76,77,78,79].

Whether low dairy consumption also leads to higher rates of bone fracture is a subject of ongoing controversy [80]. Three observational studies of adult women showed a significant increase in the risk of fracture with low dairy intake [65,75,81,82]. One population-based study of 601 Finnish elderly adults, which defined lactose intolerance on the basis of a single nucleotide polymorphism of the lactase gene (CC genotype) associated with reduced dairy intake, found that those with CC genotype had over 3-fold increase in the odds of hip fracture and nearly a 2-fold increase in the odds of wrist fracture compared to the TT genotype [83]. This association did not hold true in the Finnish postmenopausal women [84]; however, women in the maldigester group were more frequent users of calcium supplements. Since the publication of the National Institutes of Health (NIH) Consensus Statement on Lactose Intolerance in 2010 [65,85], a meta-analysis of data from 12 cohort studies reported no overall association between milk intake and hip fractures [86] and two subsequent cohort studies reported either no association in elderly Europeans [87] or a non-significant lower risk in U.S. men and women [88]. However, a recently published analysis of two U.S. cohorts, which included 123,906 elderly men and women followed for up to 32 years, found that each additional serving of milk per day was associated with a significant 8% lower risk of hip fracture, which was not explained by the calcium, vitamin D, or the protein content of milk [85]. These findings suggest that the association between dairy intake and bone fracture risk is positive but relatively small and dependent on factors that vary with the population of interest.

With regard to the fracture risk in children, two studies reported that children who avoid milk for more than 4 months had increased risk of bone fractures [44,62,65]. This effect may persist into adulthood as a retrospective study of postmenopausal women in NHANES III found that low milk intake during childhood doubled the risk of fracture in later years [89]. However, a more recently published cohort study of 96,000 postmenopausal women and elderly men found no association between milk consumption during teenage years and subsequent risk of hip fracture [90].

## 8. Recommendations for Dairy Consumption by Public Health Organizations

The 2015–2020 Dietary Guidelines for Americans [91] recommend consuming 3 servings of dairy per day, regardless of an individual’s ability to digest lactose, due to several important considerations emphasized in the 2010 NIH Consensus Statement on Lactose Intolerance [65]:

### 8.1. The Majority of People with Lactose Malabsorption Do Not Have Clinical Lactose Intolerance

Researchers estimate that approximately one-third to one-fifth of individuals with limited lactase activity will have digestive symptoms [92]. Nicklas et al. [93,94] reported that in a nationally representative sample of milk consumers, only 12–13% of adults reported symptoms, despite a substantially higher prevalence of lactose nonpersistence in this population [93]. Moreover, many of the individuals who do report symptoms show no evidence of lactose malabsorption when tested using objective methods. Suarez et al. [95] measured gastrointestinal symptoms in 30 subjects who described themselves as lactose intolerant to very small amounts of milk. Of these subjects, 21 were lactose maldigesters based on breath hydrogen test of 15 g of lactose, while 9 were lactose digesters. These findings suggest that people who identify themselves as lactose-intolerant may mistakenly attribute abdominal symptoms to milk intolerance. The NIH consensus panel encouraged health practitioners to diagnose their patients using standardized testing, such as the hydrogen breath test [65].

### 8.2. Lactose Intolerance Need Not Be an Obstacle to Meeting the Need for Calcium with 3 Servings of Milk and Dairy Products

Most adults and adolescents with limited lactose digestion can tolerate at least 12 g of lactose, the amount of lactose normally found in 1 cup (240 mL) of milk [65]. The corresponding level of tolerance has not been defined for children. However, most researchers agree that amounts <6 g do not elicit symptoms. Furthermore, research findings suggest that larger amounts may be consumed by all ages if introduced gradually over a period of 2–3 weeks, due to colonic adaptation. Repeated consumption of lactose favors the development of colonic bacteria that can digest lactose for the host which diminished the symptoms [93,96]. When building lactose tolerance, it is also recommended to consume milk with other foods or as part of a meal, which slows down gastric emptying, reduces the load of lactose that enters the intestine at any time, and allows more time for bacteria to digest lactose [93]. Another dietary strategy to achieve the recommended 3 serving of dairy is to select aged cheese, which is essentially lactose free, and yogurt with active cultures, which aid in lactose digestion [97]. Dietary Guidelines for Americans also recommend choosing lactose-reduced or lactose-free milk and fortified soy beverages (soymilk) as alternatives [91].

### 8.3. Dairy Avoidance May Lead to Deficiencies of Calcium, Vitamin D, and Other Nutrients that Track with Calcium, Which May Predispose Individuals to Decreased Bone Accrual, Osteoporosis, and Other Adverse Health Outcomes

Low dairy diets are frequently inadequate in calcium and a wide spectrum of other nutrients present in milk. Dairy foods contribute 72% of calcium, 26% of riboflavin, 16% of vitamin A, 20% of vitamin B12, 18% of potassium, 16% of zinc, 15% of magnesium, and 19% of high-quality protein available for consumption in the United States [97,98]. A number of studies have indicated that excluding dairy from diet is associated with nutritional deficiencies and reduced BMD [71,72,73,93]. On the other hand, adequate intake of dairy is a marker of high dietary quality [93,99] and a correlate of lower risks of osteoporosis, hypertension, diabetes, colorectal cancer, and weight gain [93,98,100,101,102,103].

The importance of dairy intake has been promoted in several reports, including the NAM report on School Meals and Women, Infants, and Children Supplemental Food Program (WIC), the consensus statement of the National Medical Association, and the statement of American Association of Pediatrics [104,105,106,107]. The National School Lunch and the WIC Programs require providing milk in the WIC food packages and as part of school lunch, unless requested otherwise in a written statement from a child’s physician or parent [97]. Three daily servings of dairy are also part of the dietary strategy outlined in the Dietary Approaches to Stop Hypertension diet, which forms the backbone of dietary recommendations put forward by the American Heart Association and NIH’s National Heart, Lung, and Blood Institute [97].

## 9. Recommendations for Different Age and Racial/Ethnic Groups

The recommendation to consume 3 servings of dairy per day extends to all age and racial/ethnic groups, regardless of the prevalence of lactose intolerance in these groups. Children and adolescents with maldigestion should especially be encouraged to maintain dairy food intake to meet the needs of skeletal growth and to optimize peak bone mass, most of which is attained before the age of 16 [108]. Children who avoid milk ingest less-than-recommended amounts of calcium and may be at increased risk for deficient bone accretion [78,102]. Cases of severe rickets due to vitamin D deficiency have been described in children who do not consume dairy [109]. In their 2006 statement on lactose intolerance in infants, children, and adolescents, the American Association of Pediatrics stated that restriction of milk and other dairy products is not usually necessary given the available approaches to lactose intolerance [106]. Portions of 4–8 oz. (120–240 mL) spaced throughout the day and consumed with other foods may be tolerated with no symptoms by children.

Dairy consumption should also be maintained during adulthood and in the elderly. The prevalence of lactose malabsorption increases with age but symptoms of intolerance reported by individuals with malabsorption decrease over time [97]. Therefore, the elderly should be encouraged to consume dairy foods, especially given their reduced capacity to absorb calcium, to protect them from nutritional deficiencies and to reduce age-related bone loss [108].

The choice not to exclude children and racial or ethnic minorities from the general 3-serving dairy recommendation was motivated by the high risk of nutrient deficiencies in these groups. In its 2013 consensus statement, the National Medical Association (NMA), which represents the interests of African-American physicians and their patients, urged its constituents to follow the same guidelines as the general public, i.e., (1) consuming small portions of milk, (2) consuming milk with other foods, (3) opting for aged cheese and active culture yogurt, (4) choosing lactose-free or lactose-reduced milk as cow’s milk alternatives, and (5) using lactose digestive aids when needed [105]. These recommendations were published despite the well documented high prevalence of lactose intolerance among African-Americans (75–90%), Native Americans (100%), and Asian Americans (80–90%) compared to Caucasians (12%) [65,107]. The NMA rationalized their opinion stating that the diets of low-income African-Americans are more likely to be deficient in key nutrients that are easily sourced from milk and dairy and that low intake of these nutrients predisposes Blacks and Hispanics to a higher risk of chronic diseases. Similar to the NIH consensus panel, the NMA encouraged practitioners to educate their patients about the benefits of consuming the recommended 3 servings/day of dairy and to be formally tested for lactose intolerance, as symptoms may have different origins. The European Food Safety Authority Panel on Dietetic Products, Nutrition, and Allergies also emphasized the need for objective testing before recommending a low-lactose diet and the intake of lactose-reduced and lactose-free products to avoid calcium, vitamin D, and riboflavin deficiencies [110].

The NIH consensus panel acknowledged that the task of developing a general, evidence-based recommendation for the American public is difficult due to individual differences among lactose maldigesters that include absorptive capacity, intestinal transit times, gut microbiome composition, and sensitivity to symptoms [65]. Difficulty also arises from the heterogeneity of the available research and lack of a standardized method for diagnosing and reporting symptoms, especially in double-blind randomized controlled trials. Therefore, the NIH recommendation applies only to those with proven lactose malabsorption and perceived lactose intolerance, and not, for example, those with allergy to cow’s milk protein. The NIH panel stressed the urgent need for future studies to investigate the association between dairy intake and health outcomes in people with lactose intolerance, especially children, the elderly, members of racial and ethnic subgroups, and those with susceptible genetic polymorphisms.

## 10. Conclusions

Lactose intolerance has little direct influence on the absorbability of calcium in adult men and women, although additional studies in different age groups are required to reach a firm conclusion. Lactose intolerance may predispose individuals to suboptimal bone health, osteoporosis, and fragility fractures when accompanied by decreased intake or avoidance of dairy. The current recommendation for individuals with diagnosed lactose maldigestion is to consume 3 servings of dairy per day together with other foods and to opt for dairy products low in lactose, such as aged cheese and cultured yogurt. Knowing whether cow’s milk can be safely replaced from a diet by plant-based beverages requires further research to characterize their adequacy in meeting nutrient requirements and the effects of their long-term consumption on bone health.

## Figures and Tables

**Figure 1 nutrients-11-00718-f001:**
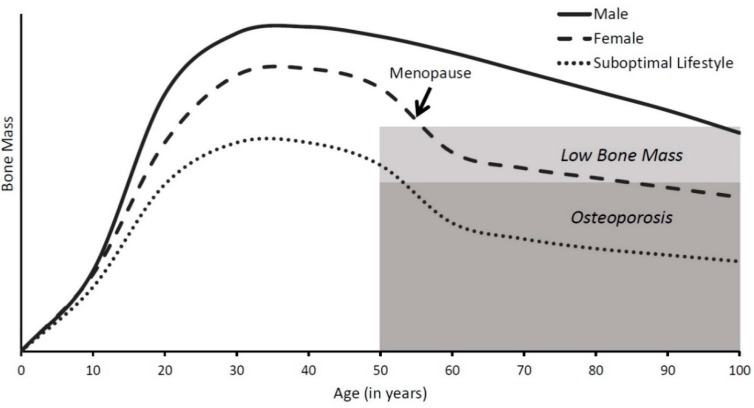
Bone mass throughout the life span. Bone accumulates rapidly in childhood and grows at maximal rates during puberty. Peak bone mass is reached by age 30, and then slowly decreases. Females typically accumulate less bone mass than males and rapidly lose bone during menopause, making them more susceptible to osteoporosis later in life.

**Table 1 nutrients-11-00718-t001:** Calcium recommendations for Americans (mg/d) [12].

Age	RDA ^1^		UL ^2^
	Female	Male	
1–3 y	700	700	2500
4–8 y	1000	1000	2500
9–18 y	1300	1300	3000
19–50 y	1000	1000	2500
51–70 y	1200	1000	2000
>70 y	1200	1200	2000

^1^ RDA = Recommended Dietary Allowance; ^2^ UL = Upper Limit.

**Table 2 nutrients-11-00718-t002:** Calcium content of dairy products and dairy substitutes [18].

Food	mg Ca/100 g	mg Ca/Serving
*Milk*		
Skim	122	299
1%	125	305
2%	120	293
Whole	113	276
*Yogurt*		
Greek	100	150
Plain	121	275
Vanilla	171	291
*Cheese*		
American	789	150
Cheddar	893	250
Colby	685	194
Cottage Cheese	111	125
Goat	140	40
Mozzarella	505	143
Provolone	756	214
Swiss	890	252
*Snacks*		
Milk Chocolate	189	47
Vanilla Ice Cream	128	84
*Dairy Substitutes*		
Almond Milk ^1^	188	451
Cashew Milk ^1^	127	451
Coconut Milk ^1^	188	451
Hemp Milk ^1^	208	499
Rice Milk ^1^	118	283
Soy Milk ^1^	123	299
Tofu Yogurt	118	309

^1^ Product fortified with calcium.

**Table 3 nutrients-11-00718-t003:** Non-dairy sources of calcium in the diet [18].

Food	mg Ca/100 g	mg Ca/Serving
*Fruits and Vegetables*		
Arugula	160	16
Beet Greens	117	44
Broccoli	47	43
Brussel Sprouts	26	25
Cabbage, Bok Choy	105	74
Cabbage, Chinese	77	59
Cabbage, Green	40	36
Cauliflower	22	24
Collards	232	84
Kale	254	53
Kohlrabi	24	32
Mustard Greens	115	64
Orange Juice ^1^	140	349
Spinach	99	30
Tofu, nigari ^2^	345	421
Tofu, plain	31	26
Turnip Greens	190	104
*Meats and Fish*		
Beef	18	20
Canned Pink Salmon	215	183
Canned Sockeye Salmon	198	168
Herring	83	71
Oyster	44	37
Trout	43	37
Turkey	19	21
Walleye	110	175
*Nuts, Beans, and Seeds*		
Almonds	269	76
Beans, Black	105	40
Beans, Pinto	111	40
Beans, Red	130	60
Beans, Soya	62	81
Beans, White	49	64
Hazelnuts	114	32
Sesame Seeds	975	88
Sunflower Seeds	78	36

^1^ Product fortified with calcium. ^2^ Nigari is mineral enriched source of calcium used to make tofu.

**Table 4 nutrients-11-00718-t004:** Nutrient composition of test beverages [18].

	Water	Cow’s Milk		Dairy Substitute Beverages			
	Bottled	Skim ^1^	Whole ^2^	Soy ^3^	Rice ^3^	Almond ^3^	Coconut ^3^
*Energy (kcal)*	0	81	149	104	113	91	70
*Macronutrients (g)*							
Protein	0	8.1	7.7	6.3	0.7	1.0	0
Fat	0	0.2	7.9	3.6	2.3	2.5	4.5
Carbohydrates	0	12	12	12	22	16	8
*Minerals (mg)*							
Calcium	24	293	276	299	283	451	101
Magnesium	5	26	24	36	26	17	41
Phosphorus	0	242	205	104	134	19	-
Potassium	0	347	322	296	65	120	41
Sodium	5	101	105	114	94	151	0
*Vitamins (IU)*							
Vitamin A	0	490	395	450	499	499	499
Vitamin D	0	113	124	104	101	101	120

^1^ Fortified with vitamins A and D. ^2^ Fortified with vitamin D. ^3^ Fortified with calcium and vitamins A and D.

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
