# Peer review of "Lactose Intolerance and Bone Health: The Challenge of Ensuring Adequate Calcium Intake"

_nutrients, 2019, doi:10.3390/nu11040718_

Round 1
Reviewer 1 Report
I find this manuscript convincing in the general hypothesis that subjects requiring greater skeletal health are better advised to increase lactose tolerance to the point where they can ingest three servings of dairy products daily than for them to avoid dairy products and use non-dairy alternatives that have not been well tested as to there efficacy. Well done!
Author Response
We sincerely appreciate the reviewer’s positive and constructive feedback. We have revised the manuscript according to reviewers’ comments. All changes were highlighted or marked by tracking in the revised version. Below are responses to all reviewers’ comments. The resubmission has an added list of abbreviations following the conclusion on page 11.
Reviewer #1:
Comment: I find this manuscript convincing in the general hypothesis that subjects requiring greater skeletal health are better advised to increase lactose tolerance to the point where they can ingest three servings of dairy products daily than for them to avoid dairy products and use non-dairy alternatives that have not been well tested as to the efficacy. Well done!
Response: Most sincere thanks for a positive feedback.
Reviewer 2 Report
Lactose intolerance and adequate calcium intake are skeletally relevant aspects that will be of interest to readers; factors potentially contributing to bone fragility are of ongoing concern.
The title is clear and concise and the Abstract and Conclusions are succinct.
It is a comprehensive review supported by more than 100 references representative of major authors in the field of bone loss and osteoporosis.
The density of facts in the text is balanced by essential summarising figures and tables and the overall structure is logical and well-defined by appropriate headings.
It is fluent, written with assurance and packed with useful information.
There are a number of minor editorial points for consideration as follows:
Line 9 ......impacts upon many......
Lines 29 & 33 .........enormous....... .........devastating....... These are emotive words best avoided in scientific description: more moderate and effective alternative adjectives include .....major..... ....extensive....
Line 107. Table heading Dairy should be lower-case dairy.
Line 109. Similarly in heading Calcium should be calcium.
References. Occasionally the title of a journal reference uses the upper case for each word e.g references 4 and 36. Also ref 4 Baldan et al seems to be identical to ref 71 Baldan et al.
Overall this is a valuable review.
Author Response
We sincerely appreciate the reviewer’s positive and constructive feedback. We have revised the manuscript according to reviewers’ comments. All changes were highlighted or marked by tracking in the revised version. Below are responses to all reviewers’ comments. The resubmission has an added list of abbreviations following the conclusion on page 11.
Reviewer #2:
Comment: Lactose intolerance and adequate calcium intake are skeletally relevant aspects that will be of interest to readers; factors potentially contributing to bone fragility are of ongoing concern. The title is clear and concise and the Abstract and Conclusions are succinct.
Comment: It is a comprehensive review supported by more than 100 references representative of major authors in the field of bone loss and osteoporosis.
Response: Thank you. We tried to capture the topic from a wide perspective and address the interests of both the researchers and the practitioners in the area.
Comment: The density of facts in the text is balanced by essential summarizing figures and tables and the overall structure is logical and well-defined by appropriate headings.
Comment: It is fluent, written with assurance and packed with useful information.
Comment: There are a number of minor editorial points for consideration as follows:
Comment: Line 9 ......impacts upon many......
Response: This has been fixed. Thank you.
Comment: Lines 29 & 33 .........enormous....... .........devastating....... These are emotive words best avoided in scientific description: more moderate and effective alternative adjectives include .....major..... ....extensive....
Response: Poor word choice has been corrected. Thank you for instructive comments.
Comment: Line 107. Table heading Dairy should be lower-case dairy.
Response: Revised. Thank you.
Comment: Line 109. Similarly in heading Calcium should be calcium.
Response: Amended. Thank you.
Comment: References. Occasionally the title of a journal reference uses the upper case for each word e.g references 4 and 36. Also ref 4 Baldan et al seems to be identical to ref 71 Baldan et al.
Response: The upper cases were replaced with lower cases where appropriate and reference #71, correctly identified by the reviewer as a duplicate of #4, was deleted. Tremendous thanks for a throughout revision.
Comment: Overall this is a valuable review.